# Early death after a diagnosis of metastatic solid cancer–raising awareness and identifying risk factors from the SEER database

**Opher Globus**[1,2☯], **Shira Sagie**[1,3,4☯], **Noy Lavine**[5], **Daniel Itshak Barchana**[5],
**Damien Urban**[1,2]*

1 Institute of Oncology, Sheba Medical Center, Ramat Gan, Israel, 2 Sackler Faculty of Medicine, Tel Aviv University, Tel Aviv, Israel, 3 The Sheba Talpiot Medical Leadership Program, Israel, 4 Department of Molecular Cell Biology, Weizmann Institute of Science, Rehovot, Israel, 5 St. George's University of London Medical Program Delivered by University of Nicosia Medical School, Nicosia, Cyprus

☯ These authors contributed equally to this work.
* Damien.Urban@sheba.health.gov.il

**Data Availability Statement:** All relevant data are within the paper and its Supporting Information files.

## Abstract

### Background

Cancer death rates are declining, in part due to smoking cessation, better detection and new treatments; nevertheless, a large fraction of metastatic cancer patients die soon after diagnosis. Few studies and interventions focus on these patients. Our study aims to characterize early mortality in a wide range of metastatic solid tumors.

### Methods

We retrieved data on adult patients diagnosed with pathologically confirmed *de- novo* metastatic solid tumors between the years 2004–2016 from the Surveillance, Epidemiology, and End Results database (SEER). Our primary outcome was cancer specific early death rate (defined as death within two months of diagnosis). Additional data extracted included socio-demographical data, tumor primary, sites of metastases, and cause of death.

### Results

109,207 (20.8%) patients died of *de-novo* metastatic cancer within two months of diagnosis. The highest rates of early death were found in hepatic (36%), pancreato-biliary (31%) and lung (25%) primaries. Factors associated with early death included primary site, liver, and brain metastases, increasing age, and lower income. Cancer was the cause of death in 92.1% of all early deaths. Two-month mortality rates have moderately improved during the study period (from 22.4% in 2004 to 18.8% in 2016).

### Conclusion

A fifth of *de-novo* metastatic cancer patients die soon after diagnosis, with little improvement over the last decade. Further research is required to better classify and identify patients at

**Funding:** The authors received no specific funding for this work.

**Competing interests:** Dr Urban reports receiving consulting fees from Merck, Sharpe & Dohme, Roche Israel, Takeda, Nucleai, Rhenium Oncotest, and lecture fees from Astrazeneca, Merck, Sharpe & Dohme, Roche, Takeda, Bristol Myers Squibb, and Merck Seronoe. Dr Globus reports receiving lecture fees from Pfizer Lilly, Roche, Astra Zeneca, Novartis, Gilead, and Merck, Sharpe & Dohme, consulting fees from Lilly, Gilead and Novartis, and expenses for conferences from Pfizer, Medison, Rhenium Onctotest, and Gilead. Dr Sagie, Levina, and Brachana report no potential conflicting interests. This does not alter our adherence to PLOS ONE policies on sharing data and materials.

risk for early mortality, which patients might benefit from faster diagnostic tracks, and which might avoid invasive and futile diagnostic procedures.

## Introduction

Despite improvements in early detection and treatment, cancer remains one of the leading causes of death in the United States and around the world [1, 2]. Major advances in the personalization of systemic treatment of cancer have demonstrated rapid responses and long-term survival in some patients, even in the metastatic setting [3]. However, clinicians, often encounter patients with suspected *de-novo* metastatic disease who are extremely ill at the time of presentation. In these clinical situations, physicians, patients, and families are faced with fundamental decisions. On the one hand, particularly due to advances in personalized oncology, some select patients can rapidly improve with targeted treatments. On the other hand, diagnostic procedures are not without financial costs and potential complications [4], often resulting in hospital admissions and poor end-of-life care. As many of the pivotal improvements require predictive biomarkers, which can take weeks to return, the benefit of these treatment advances may elude many patients.

The aim of our current study is to describe early mortality in solid tumors and to characterize which patients with *de-novo* metastatic disease die within two months of diagnosis. As early mortality from cancer is due to a combination of complex tumor-related, patient-related, and healthcare-related mechanisms, our objective is to present and disseminate the scope of this phenomenon to healthcare professionals involved in the diagnosis and treatment of cancer and to identify characteristics associated with early mortality, thereby helping guide physicians, patients, and families in appropriate clinical decisions.

## Methods

### Study population and design

Patients with histologically confirmed *de-novo* metastatic cancer were identified from the Surveillance, Epidemiology, and End Results (SEER) database from 2004 to 2016.

The SEER database, maintained by the NCI in the U.S., is a comprehensive and publicly accessible cancer database. It offers detailed information on cancer incidence, prevalence, survival, and treatment patterns. Since 1973, SEER facilitates research, policymaking, and clinical decision-making by providing valuable insights into cancer patterns and outcomes. The 18 population based SEER cancer registries cover 28% of the US population [5]. Given the nature of this study, there was no requirement for institutional review board submission. Access to the SEER data was in accordance with the SEER data agreement.

Patients included were at least 18 years of age and presented with a first diagnosis of pathologically confirmed metastatic cancer. Patients with recurrent metastatic disease were excluded. As the focus of our study was metastatic solid tumors, we excluded patients with primary neurological tumors, hematological malignancies, and rare solid tumors (peripheral nervous system tumors, tumors stemming from the retroperitoneum, non-lung cancer chest tumors, tumors identified as not otherwise specified from the male genital, female genital, and the urinary tract).

### Study variables

Our primary outcome was cancer-specific two-month mortality from the date of diagnosis.

Cause-specific survival in the SEER data is a measure that focuses on the survival of individuals attributed to a specific cause of death, excluding other causes. It involves specifying the cause of death and considering individuals who die from causes other than the specified cause as censored. The cancer registries utilize algorithms to process information from death certificates and identify a single, disease-specific underlying cause of death. However, there can be challenges in accurately attributing a single cause of death, leading to potential misattribution. To address this, SEER utilizes cause-specific death classification variables that consider tumor sequence, the site of the original cancer diagnosis, and comorbidities to capture deaths related to the specific cancer even if not explicitly coded as such. This approach leads to a relatively small bias [6].

Additional data extracted included age, sex, race, residence, income, tumor primary site, metastasis location (available only after 2010 for liver, bone, brain, and lung), year of diagnosis (grouped in four time periods 2004–2006, 2007–2009, 2010–2012, 2013–2016), cause of death, and survival by months.

We separately analyzed twelve groups of tumor locations: bladder, breast, colorectal, gastro-esophageal, hepatocellular, head and neck, kidney, lung, melanoma, ovary, pancreato-biliary, and prostate, all defined by primary tumor location except for melanoma which was defined by both histology and location.

## Statistical analysis

The Mann-Whitney $U$ test was used for variables that did not follow a normal distribution and the chi square test and Fisher Exact test were used as appropriate for categorical variables. We fitted multivariable logistic models. Data for metastasis sites was available since 2010, thus we fitted separate multivariable models for the whole population and for patients diagnosed after 2010. Adjusted OR and 95% confidence intervals (CI) were obtained for the independent variables from these models. $P < 0.05$ was considered statistically significant. The statistical analyses were performed using R version 4.0.3.

## Results

### Baseline characteristics

We identified 525,780 patients diagnosed with *de-novo* metastatic solid malignancies between 2004 and 2016 of which 109,207 (20.77%) died of cancer within two months of diagnosis. Patient characteristics are presented in Table 1. Most patients were male (52.1%), white (78.6%), and 50–70 years old (51.7%). The most common primary site was lung (40%) followed by pancreato-biliary (11.2%) and colorectal cancer (10.8%). Patients that died within two months compared to those who did not, were older (median 70 vs 65), more frequently male (54.3% vs 51.5%), had lower income (56.8% vs 54.9%) and more frequently had brain (15% vs 12.9%), lung (49.2% vs 33.5%) and liver metastasis (33.5% vs 28%).

Breast and prostate cancer primaries were less frequent in the early mortality group (3% and 0.9%, respectively); in contrast, lung and pancreato-biliary primaries were more frequent (48.6% and 16.8%, respectively). Primary tumor distribution in the whole cohort and in the early mortality group by gender appears in Fig 1.

### Causes of death

Within our cohort, 109,207 (20.8%) patients diagnosed with *de-novo* metastatic solid cancer died of cancer within two months of diagnosis. Cancer was the cause of death in 92.1% of early deaths (Fig 2A). The most common non-cancer cause of early death was heart disease

**Table 1. Characteristics of the study group classified by early mortality.**

| | Overall | Early mortality | | p |
| | | No | Yes | |
|---|---|---|---|---|
| N | 525780 | 416573 | 109207 | |
| Sex, Male (%) | 273976 (52.1) | 214661 (51.5) | 59315 (54.3) | <0.001 |
| Race (%) | | | | <0.001 |
| White | 413207 (78.6) | 325598 (78.2) | 87609 (80.2) | |
| Black | 69371 (13.2) | 55525 (13.3) | 13846 (12.7) | |
| American Indian/Alaska Native | 3576 (0.7) | 2814 (0.7) | 762 (0.7) | |
| Asian or Pacific Islander | 38627 (7.3) | 31769 (7.6) | 6858 (6.3) | |
| Unknown | 999 (0.2) | 867 (0.2) | 132 (0.1) | |
| Age (Median [IQR])) | 66 [57, 75] | 65 [56, 74] | 70 [61, 78] | <0.001 |
| Residence Category (%) | | | | <0.001 |
| Metropolitan | 313730 (59.8) | 248942 (59.8) | 64788 (59.4) | |
| Rural | 27729 (5.3) | 21788 (5.2) | 5941 (5.4) | |
| Urban | 183506 (35.0) | 145222 (34.9) | 38284 (35.1) | |
| Income below $65K/year (%) | 284945 (55.3) | 224293 (54.9) | 60652 (56.8) | <0.001 |
| Grouped Years (%) | | | | <0.001 |
| 2004–2006 | 113549 (21.6) | 88288 (21.2) | 25261 (23.1) | |
| 2007–2009 | 118303 (22.5) | 93352 (22.4) | 24951 (22.8) | |
| 2010–2012 | 119990 (22.8) | 95143 (22.8) | 24847 (22.8) | |
| 2013–2016 | 173938 (33.1) | 139790 (33.6) | 34148 (31.3) | |
| Bone Metastasis, Yes (%) | 93642 (33.0) | 75821 (33.3) | 17821 (31.7) | <0.001 |
| Brain Metastasis, Yes (%) | 37565 (13.3) | 29214 (12.9) | 8351 (15.0) | <0.001 |
| Lung Metastasis, Yes (%) | 104138 (36.6) | 76277 (33.5) | 27861 (49.2) | <0.001 |
| Liver Metastasis, Yes (%) | 81644 (29.1) | 63075 (28.0) | 18569 (33.5) | <0.001 |
| Primary Cancer Location (%) | | | | |
| Bladder | 4819 (0.9) | 3791 (0.9) | 1028 (0.9) | |
| Breast | 34387 (6.5) | 31066 (7.5) | 3321 (3.0) | |
| CRC | 56763 (10.8) | 48506 (11.6) | 8257 (7.6) | |
| Gastro-esophageal | 35563 (6.8) | 27269 (6.5) | 8294 (7.6) | |
| HCC | 7964 (1.5) | 5119 (1.2) | 2845 (2.6) | |
| Head and Neck | 7647 (1.5) | 6968 (1.7) | 679 (0.6) | |
| Kidney | 17399 (3.3) | 14803 (3.6) | 2596 (2.4) | |
| Lung | 210027 (39.9) | 156898 (37.7) | 53129 (48.6) | |
| Melanoma | 6807 (1.3) | 5755 (1.4) | 1052 (1.0) | |
| Ovary | 17854 (3.4) | 15027 (3.6) | 2827 (2.6) | |
| Pancreato-biliary | 59071 (11.2) | 40683 (9.8) | 18388 (16.8) | |
| Prostate | 28219 (5.4) | 27283 (6.5) | 936 (0.9) | |

*Income- omitted 10,615 NA's, Residence-omitted 811 NA's, Bone metastasis -omitted 242413 NA's, Brain metastasis—Omitted 244295 NA's, Lung metastasis—Omitted 245322 NA's, Liver metastasis—Omitted 241850 NA's. HCC: Hepatocellular carcinoma. CRC: Colorectal cancer. NA = Not available

n = 2,896 (0.56% of all patients) followed by infection n = 1,010 (0.19% of all patients). The ten most common non-cancer causes of early death appear in Fig 2B.

## Primary tumor specific early mortality rates

The two-month cancer specific mortality rate was highest in patients with hepatocellular (36%), pancreato-biliary (31%), lung (25%), gastro-esophageal (23%) and bladder (21%) primaries and lowest in patients with prostate (3%), head and neck (9%) and breast (10%)

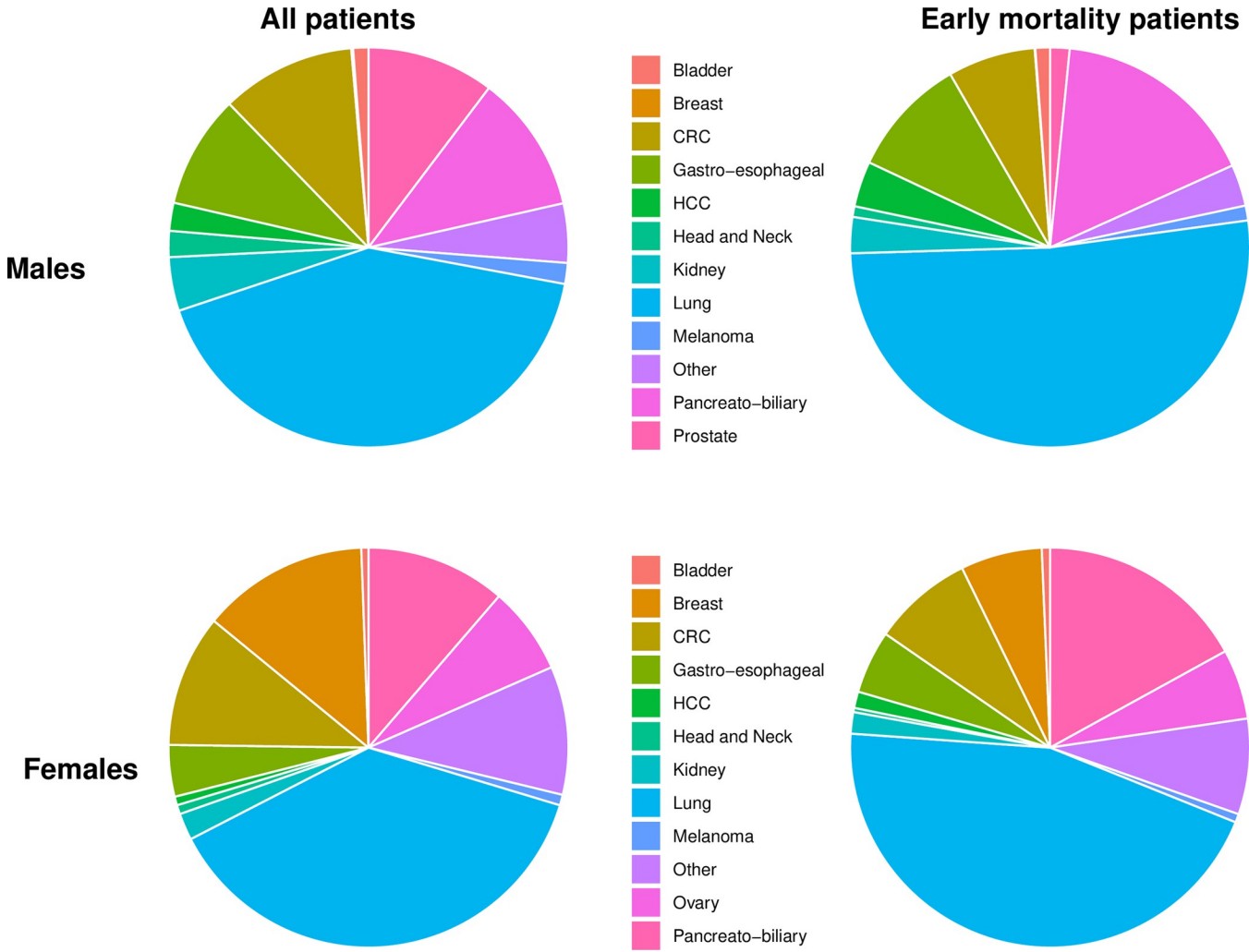

**Fig 1. Disease site distribution by gender in the whole cohort and in the early mortality group.** Left Panel-525,780 Patients with *de-novo* metastatic cancer were analyzed by disease site origin, separated by gender. Right Panel -109,207 patients which died in the first two months after disease diagnosis analyzed by disease site origin, separated by gender. Primary site color code differs between genders and appears in the middle.

primaries (Fig 3). Black patients had a significantly higher early mortality rate in breast, gastro-esophageal, pancreato-biliary, ovarian and kidney cancer, whereas white patients had a significantly higher mortality rate in lung, prostate cancer, and melanoma. These differences remain statistically significant in multivariate analysis. Fig 3 displays early mortality rates by primary tumor, race, and year of diagnosis.

In most cancer types a statistically significant reduction in early mortality was observed between 2004 and 2016, with the overall two-month cancer specific mortality rate improving during the period of our study from 22.4% in 2004 to 18.8% in 2016. The strongest relative reductions in early mortality were documented in prostate, kidney, and breast cancers. Cancers without a statistically significant reduction in early mortality include bladder cancer, head and neck cancer and melanoma (Fig 3B).

### Multivariate analysis of all cancers

On multivariate analysis, factors significantly associated with increased two-month mortality (Table 2) included site of metastases, particularly liver metastases (data starting from 2010, OR

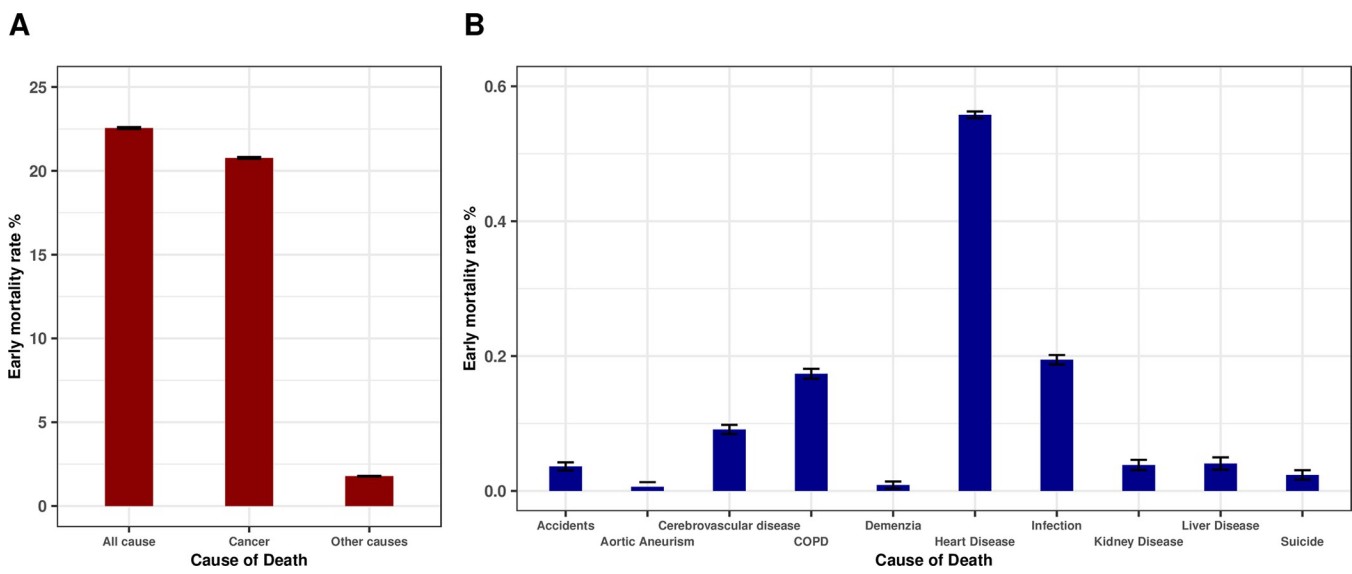

**Fig 2. Major causes of early death in de novo diagnosed metastatic cancer patients.** All cause and cancer specific early death rates (A). Ten major causes of non-cancer early death (B). Y axes present percentages in both plots.

1.91 CI 1.87–1.96, p<0.001), male sex (OR 1.17 CI 1.16–1.19, p<0.001), income below $65K a year (OR 1.12 CI 1.11.13, p<0.001), race (black and native American, OR 1.07 CI 1.04–1.09, p<0.001 and OR 1.13 CI 1.02–1.24, p<0.001 respectively), and increasing age (OR 1.03 per year p<0.001).

## Discussion

Our study reports that approximately 20% of patients with biopsy proven *de-novo* metastatic solid cancer die within two months of diagnosis. Previous studies of individual cancers have also reported high early mortality rates. An analysis of the national cancer database (NCDB) showed that between 2006 and 2014, 13% of patients with metastatic non-small cell lung cancer (NSCLC) died within 30 days of diagnosis [7]. A SEER analysis of patients with metastatic breast cancer showed that in 2013, 13.4% of patients died within one month, while a SEER analysis of patients with pancreatic adenocarcinoma showed that approximately half of patients died within two months of diagnosis [8, 9]. These studies demonstrated that early mortality is a prevalent phenomenon, associated with increased age, insurance status, geographic location, and comorbidity burden. Nevertheless, the scope of early mortality across many solid tumors and the total magnitude of this phenomenon is underrecognized.

Death so quickly after a *de-novo* metastatic solid cancer diagnosis may be due to causes related to the cancer itself or because of non-cancer related comorbidities. In our study over 90% of early mortalities (Fig 2) were due to the cancer itself, a figure significantly higher than previously reported [10]; likely because our cohort included only patients who had a histologically confirmed diagnosis, indicating they were well enough to undergo invasive testing during the diagnostic process. We chose to examine two-month mortality because this is the period often required to see an improvement from many cancer therapies, while death within this period suggests that patients are unlikely to benefit from many systemic treatments. In our study patients with pancreato-biliary, lung, and liver primaries had the highest proportion of early mortality. Patients with liver metastases were almost twice as likely to suffer early mortality independent of other factors.

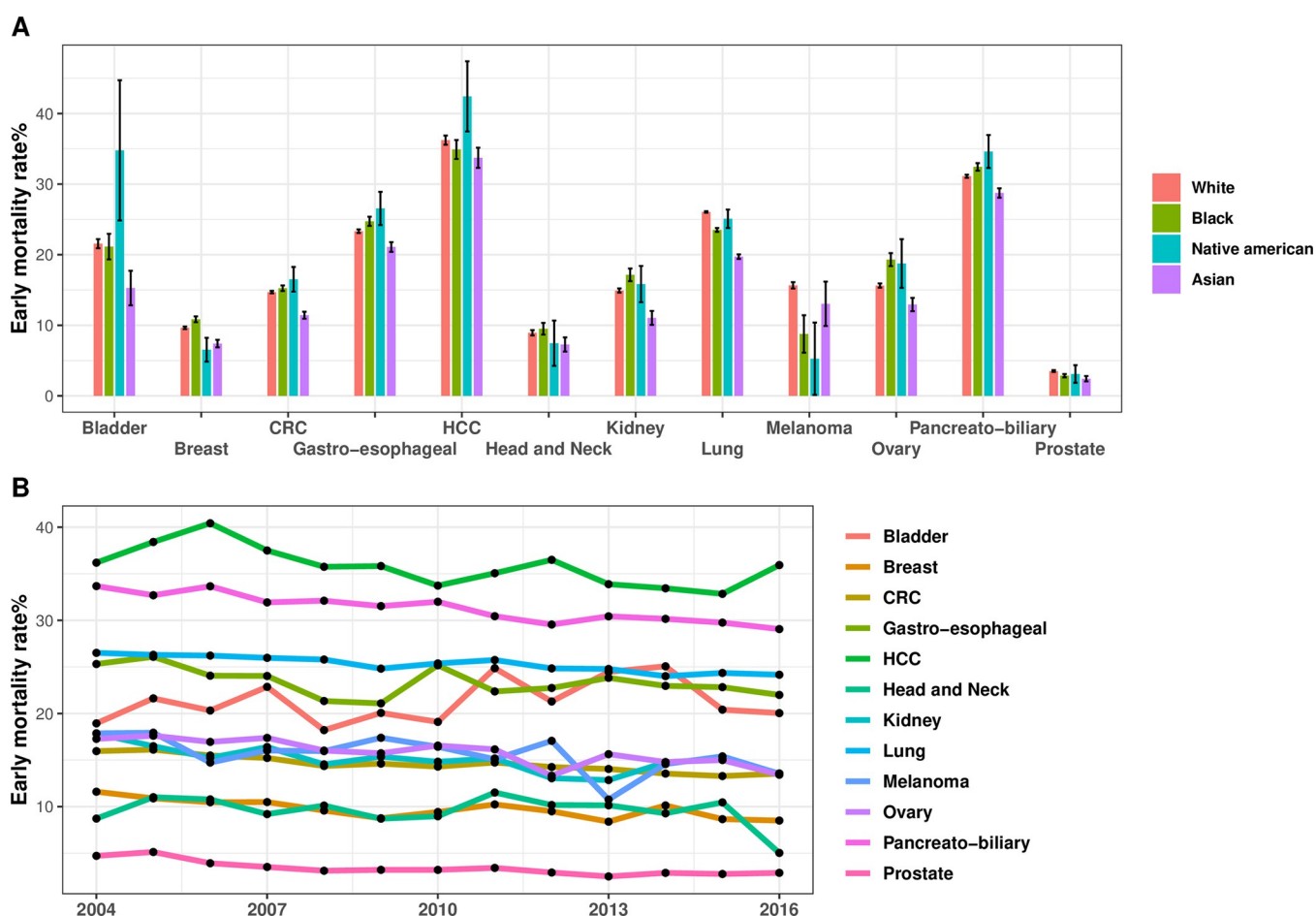

**Fig 3. Early mortality rates by primary tumor, race, and year of diagnosis.** Early mortality rates were calculated separately for each primary tumor group in each race group (A). Early mortality trends by primary tumor site and year of diagnosis (B). Y axes present percentages.

Our findings also shed light on the role of income in cancer outcomes. Lower income was identified as a significant risk factor for early mortality in *de-novo* metastatic cancer patients. From a policy perspective, this presents a crucial area for intervention. Lower income patients might face barriers to access timely diagnosis, treatment, and comprehensive care, contributing to worse outcomes. Addressing this issue requires broader policy interventions aimed at minimizing healthcare disparities and improving access and affordability of care, particularly for those most vulnerable.

The cancer diagnostic process itself is costly and time consuming, taking a median of 35 days for a diagnosis and an additional 23 days for molecular testing results in certain tumor types, such as non-small cell lung cancer [11, 12]. Unfortunately for many patients, early cancer death in inevitable, even with immediate treatment. Early identification of high-risk patients can help avoid costly evaluations and prompt consideration of end of life care, even without tissue diagnosis. Conversely, rapid treatment can yield significant responses in certain patient subsets. For instance, lung cancer accounts for nearly 50% of patients dying within 2 months of a *de-novo* metastatic diagnosis. The remarkable advances in non-small cell lung cancer targeted therapy can sometimes prompt rapid responses in up to 70% of patients. Given that 20–30% of patients have a targetable driver mutation, there lies a potential to significantly extend survival if such mutations are detected promptly. Using minimally invasive

**Table 2. Multivariable logistic regression analysis of the correlates of early mortality in all patients.**

| | Multivariable 2010+ | | Multivariable all data | |
|---|---|---|---|---|
| | OR, CI | p-value | OR, CI | p-value |
| Sex: | | | | |
| Women | | Ref. | | Ref. |
| Men | 1.19 [1.16,1.21] | <0.001 | 1.17 [1.16,1.19] | <0.001 |
| Grouped Years: | | | | |
| 2004–2006 | NA | NA | Ref. | Ref. |
| 2007–2009 | NA | NA | 0.92 [0.91,0.94] | <0.001 |
| 2010–2012 | Ref. | Ref. | 0.91 [0.89,0.93] | <0.001 |
| 2013–2016 | 0.94 [0.92,0.96] | <0.001 | 0.85 [0.84,0.87] | <0.001 |
| Race: | | | | |
| White | Ref. | Ref. | Ref. | Ref. |
| Black | 1.05 [1.02,1.08] | 0 | 1.07 [1.04,1.09] | <0.001 |
| American Indian/Alaska Native | 1.01 [0.87,1.16] | 0.46 | 1.13 [1.02,1.24] | 0.02 |
| Asian or Pacific Islander | 0.8 [0.77,0.84] | <0.001 | 0.8 [0.78,0.82] | <0.001 |
| Unknown | 0.78 [0.6,1] | <0.001 | 0.76 [0.62,0.91] | <0.001 |
| Age | 1.03 [1.03,1.03] | <0.001 | 1.03 [1.03,1.03] | <0.001 |
| Income Category: | | | | |
| Above 65K | Ref. | Ref. | Ref. | Ref. |
| Below 65K | 1.16 [1.14,1.19] | <0.001 | 1.12 [1.1,1.13] | <0.001 |
| Residence Category | | | | |
| Metropolitan | Ref. | Ref. | Ref. | Ref. |
| Rural | 0.94 [0.9,0.99] | 0.8 | 0.94 [0.91,0.98] | <0.001 |
| Urban | 0.97 [0.95,0.99] | 0.19 | 0.98 [0.96,0.99] | <0.001 |
| Bone Metastasis: | | | NA | NA |
| No | Ref. | Ref. | NA | NA |
| Yes | 1.17 [1.14,1.2] | <0.001 | NA | NA |
| Brain Metastasis: | | | NA | NA |
| No | Ref. | Ref. | NA | NA |
| Yes | 1.17 [1.14,1.21] | <0.001 | NA | NA |
| Lung Metastasis: | | | NA | NA |
| No | Ref. | Ref. | NA | NA |
| Yes | 1.29 [1.26,1.32] | <0.001 | NA | NA |
| Liver Metastasis: | | | NA | NA |
| No | Ref. | Ref. | NA | NA |
| Yes | 1.91 [1.87,1.96] | <0.001 | NA | NA |

*We generated two logistic models, the first including data after 2010 that contains information about metastatic sites, the second model includes all data since 2004.

diagnostic tools, such as liquid biopsies identifying genomic alterations [13, 14], it is possible to identify patients that have a high chance of targetable mutations and a subsequently significant response to treatments, while potentially avoiding more invasive and time-consuming diagnostic interventions such as tissue biopsies. The nuanced distinction between patients who are nearing death, for whom both diagnosis and treatment are likely to be futile versus those who could potentially benefit from treatment, carries significant clinical implications. Identifying this fine line between treatable and non-treatable conditions is of utmost importance.

Despite the substantial early mortality rate in our study, we also observed a small yet encouraging trend of improvement over time amongst most solid tumors. While the observed

trend in our analysis is likely to persist, delayed diagnosis and healthcare seeking behaviors during the COVID-19 pandemic may have a dampening effect on this positive trend [15, 16]. Looking into the future, the advent of blood based multicancer early detection assays stands to improve early mortality from *de-novo* metastatic cancer by potentially lowering costs, enhancing early detection and streamlining the diagnostic process in patients with suspected metastatic disease. Our study further emphasizes the profound implications this technology may have on cancer care [17–19].

Our study, like others utilizing the SEER database, has several limitations [20]. We do not have information on important confounders including performance status, comorbidities, socioeconomic environment, and healthcare facilities treated. In addition, we do not have data on physician referral or treatment patterns. Our cohort includes only patients with *de-novo* metastatic cancer and is not applicable to patients with metastatic recurrences.

In conclusion, our study highlights the magnitude of early mortality among *de-novo* metastatic cancer patients and identifies risk factors associated with early mortality including cancer primary, liver and brain metastases, advanced age, and lower income. We report that the principal cause of death for these patients was their metastatic disease, rather than any co-existing medical conditions. Further studies are required to better identify patients at high risk for early mortality, facilitating educated discussion with patients and caregivers. If properly identified, a unique and more efficient diagnostic pathway for this patient population may help improve the outcomes of these patients.

## Supporting information

**S1 Data.**
(CSV)

## Acknowledgments

**Disclosures:** Dr Urban reports receiving consulting fees from Merck, Sharpe & Dohme, Roche Israel, Takeda, Nuclei, Rhenium Oncotest, and lecture fees from Astrazeneca, Merck, Sharpe & Dohme, Roche, Takeda, Bristol Myers Squibb, and Merck Seronoe. Dr Globus reports receiving lecture fees from Pfizer Lilly, Roche, Astra Zeneca, Novartis, Gilead, and Merck, Sharpe & Dohme, consulting fees from Lilly, Gilead, Pfizer, and Novartis, and expenses for conferences from Pfizer, Medison, Rhenium Onctotest, and Gilead. Dr Sagie, Lavine, and Barchana report no potential conflicting interests.

## Author Contributions

**Conceptualization:** Opher Globus, Shira Sagie, Damien Urban.

**Data curation:** Opher Globus, Shira Sagie, Noy Lavine, Daniel Itshak Barchana, Damien Urban.

**Formal analysis:** Shira Sagie, Noy Lavine, Daniel Itshak Barchana, Damien Urban.

**Investigation:** Opher Globus, Shira Sagie, Damien Urban.

**Methodology:** Opher Globus, Shira Sagie, Damien Urban.

**Project administration:** Opher Globus, Shira Sagie, Damien Urban.

**Resources:** Shira Sagie, Damien Urban.

**Software:** Shira Sagie.

**Supervision:** Opher Globus, Damien Urban.

**Validation:** Opher Globus, Shira Sagie, Damien Urban.

**Visualization:** Opher Globus, Shira Sagie, Damien Urban.

**Writing – original draft:** Opher Globus, Shira Sagie, Damien Urban.

**Writing – review & editing:** Opher Globus, Shira Sagie, Noy Lavine, Daniel Itshak Barchana, Damien Urban.

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
