## [Decision Letter · Decision Letter 0]

26 May 2023

PONE-D-23-02061Early death after a diagnosis of metastatic solid cancer – raising awareness and identifying risk factors from the SEER database.PLOS ONE

Dear Dr. Urban,

Thank you for submitting your manuscript to PLOS ONE. After careful consideration, we feel that it has merit but does not fully meet PLOS ONE’s publication criteria as it currently stands. Therefore, we invite you to submit a revised version of the manuscript that addresses the points raised during the review process.

We have completed the review process for your manuscript titled "Early death after a diagnosis of metastatic solid cancer – raising awareness and identifying risk factors from the SEER database." (Manuscript ID: PONE-D-23-02061), and I would like to thank you for submitting your work to PLOS ONE.

Your manuscript was reviewed by two independent experts in the field. I have carefully considered the reviewers' feedback, which provided valuable insights into the strengths and weaknesses of your work. The reviewers' comments have highlighted both the potential for major revisions and concerns leading to a recommendation for rejection. I will summarize the reviewers' evaluations and provide guidance regarding the next steps for your manuscript.

Reviewer 1:

Reviewer 1 has provided a thorough review and identified several significant issues that need to be addressed. The reviewer has pointed out areas where clarification, further analysis, or additional experiments are necessary to strengthen the paper's claims. The reviewer's suggestions and constructive criticism offer a clear path towards improving the manuscript. I highly encourage you to carefully consider these comments and implement the proposed revisions in order to enhance the scientific rigor and overall quality of your work.

Reviewer 2:

Reviewer 2, on the other hand, expressed reservations about the manuscript and recommended rejection. While their feedback may appear discouraging, it is important to note that scientific opinions can vary, and reviewer evaluations do not always align. However, their concerns cannot be overlooked, and it is essential to carefully address them in any revision attempt. Please take their comments into consideration and attempt to address their main concerns to the best of your ability.

Considering the time in which the paper was under review, and the varying evaluations from the reviewers, I propose that you make a broad and substantial modification of the paper, in accordance with the comments of both reviewers. By addressing their concerns, you will have the opportunity to strengthen your work and increase its chances of acceptance.

It is important to emphasize that despite your revisions, we cannot guarantee that the paper will be finally accepted. However, by incorporating the suggested changes and responding adequately to the reviewers' comments, you will greatly improve the manuscript's chances of a favorable outcome.

I kindly request that you submit a revised version of your manuscript, which should address all the concerns raised by the reviewers. In your resubmission, please include a detailed response letter that outlines how you have addressed each comment, providing clear references to the corresponding revisions in the manuscript. This will assist the reviewers and the editorial team in evaluating the improvements made.

Should you decide to submit a revised version, please adhere to the journal's guidelines and ensure that all required elements, such as figures, tables, and supplementary materials, are appropriately included.

I understand that major revisions can be challenging, but I encourage you to view this process as an opportunity for growth and enhancement of your work. If you require any assistance or have any questions, please do not hesitate to reach out to us.

Once again, I appreciate your dedication and the significant effort you have put into your research. I look forward to receiving your revised manuscript.

We look forward to receiving your revised manuscript.

Kind regards,

Rita De Sanctis

Academic Editor

PLOS ONE

Journal Requirements:

Dr Urban reports receiving consulting fees from Merck, Sharpe & Dohme, Roche Israel, Takeda, Nucleai, Rhenium Oncotest, and lecture fees from Astrazeneca, Merck, Sharpe & Dohme, Roche, Takeda, Bristol Myers Squibb, and Merck Seronoe. Dr Globus reports receiving lecture fees from Pfizer Lilly, Roche, Astra Zeneca, Novartis, Gilead, and Merck, Sharpe & Dohme, consulting fees from Lilly, Gilead and Novartis, and expenses for conferences from Pfizer, Medison, Rhenium Onctotest, and Gilead. Dr Sagie, Levina, and Brachana report no potential conflicting interests.   

We note that you received funding from a commercial source: [Name of Company]

Reviewers' comments:

Reviewer's Responses to Questions

**Comments to the Author**

1. Is the manuscript technically sound, and do the data support the conclusions?

Reviewer #1: Yes

Reviewer #2: No

2. Has the statistical analysis been performed appropriately and rigorously? 

Reviewer #1: Yes

Reviewer #2: No

3. Have the authors made all data underlying the findings in their manuscript fully available?

Reviewer #1: Yes

Reviewer #2: Yes

4. Is the manuscript presented in an intelligible fashion and written in standard English?

Reviewer #1: Yes

Reviewer #2: No

5. Review Comments to the Author

Reviewer #1: The Authors present the results of a very interesting analysis of the SEER database. The methods are sound and the results - the main of which is, one fifth of de novo diagnosed cancer patients die withih the first two months since diagnosis - deserve dissemination.

I can offer the following comments:

- the analysis spans from 2004 to 2016. Twelve years in which the early death rate decreased by 4%. Which are Authors' expectations for the last 6 years, 2017-2023, taking into account also the pandemic period?

- Table 1, the results do not match with the text (e.g., 54.3% of those with early dying appear to be women, but the same figures was reported for men in the text). In the same Table, what does the p-value for race refer to? there seem to be major differences in terms of ethnicities in the overall population, but no in the early vs non early mortalty groups (e.g., 80.2% of patients with early morality was white, compared with 78.2% in those without early mortality - a difference unlikely to be significant). Can the Authors carefully reconsider all data?

- the Authors should always clarify that their findings apply to patients with de novo diagnosed metastatic solid cancer (see for instance the first paragraph of the Discussion, this is not clear).

Minor:

- Why does the Introduction start with a word on cancer death rate in the US? (the Authors are from Israel and Cyprus). Can this be generalized?

- Can the Authors expand on the SEER database?

- in the sentence (Results section) 'Patients that died early were older (70 vs 65), more frequently male (54.3% vs 51.5%), had lower income (56.8% vs 54.9% had an income below $65K a year) and more frequently had brain (15% vs 12.9%), lung (49.2% vs 33.5%) and liver metastasis (33.5% vs 28%).', do the Authors mean 'than those who died after two months since diagnosis'?

- Quality of figures should be improved.

Reviewer #2: This a retrospective study based on available SEER data, investigating factors of early mortality (2-months) in solid tumors. As a general comment, I believe that the clinical question regarding the drivers of early mortality is crucial, but it is extremely difficult to assess in a cohort of patients with different solid tumors. Indeed, the biology, the response to treatments, the diagnostic process as well as symptoms and possible complication cancer and treatment related significantly differ across cancer subtypes. Therefore, it is difficult to draw generalized conclusions for all solid tumors. In addition, treatment information is missing, and as well as adverse events that may have a role in early mortality. Another missing element that can affect mortality is comorbidities status: patients with a compromised general status or organ impairment due to other diseases are expected to be more likely to rapid decline. An example is COPD in patients with lung cancer: I expect that a patient with an obstructive lung disease and lung cancer is more likely to develop respiratory failure than a patient with normal respiratory function at the time of the cancer diagnosis. In addition, the choice of cut-off of 2 months have created an intrinsic selection bias for aggressive cancer more likely to develop rapid progression. Indeed, authors in the discussion states that 2 months is the usual cut-off to see response in patients in solid tumors. If the baseline is date of biopsy as it seems, this means that patients had the time to probably received 1 cycle of therapy (in average, 1-2 weeks for pathology report, 3-4 weeks to have genomic profile/biological characterization of the tumor). With 2 months as cut-off, many patients with potentially early death are missed. Ideally, for accuracy, the cut-off for early death should be defined according to the cancer subtype.

Another issue difficult to overcome in this type of analyses is the following: how is cancer-related death defined? This information should be included in the methods since it is the main objective of the study. In cancer patients, death can be associated with different scenarios including organ failure due to cancer invasion, or secondary consequences of the cancer dissemination (i.e. hypercoagulative state, compression issues etc..). This element is difficult to retrieve from SEER published data.

Authors specified that they included only metastatic de novo, but they do not explain why. This choice introduces another selection bias: there is a potential underestimation of early death in those cancer subtype who are less likely to present as de novo (for example, head and neck cancer, that are actually expected to progress rapidly and be lethal for loco-regional complications).

The results are consistent with what expected: higher prevalence of pancreatic cancer and lung cancer, presence of visceral metastases, advanced age, and lower income were associated with early death (the latter element, very relevant form policy standpoint and has not been properly commented in the discussion). In the discussion section, it is not clear what is the main finding (new finding) and comments on clinical implications are questionable. In my opinion, palliative support needs to be activated in all patients with symptoms, in parallel to the active treatment regardless of if patient it is at risk of early death. I provide a similar argument for invasive diagnostic procedure: for the “primum non nocere” principle, less invasive procedures should be considered for all patients. Liquid biopsy can be diagnostic only for some cancer subtype (i.e. colon cancer, lung cancer) and it is not validated for all type of solid tumors, and in those case in which is validated, it has been performed as standard of care and routinely at the diagnosis for all patients. On the other hand, if a patient is considered ineligible for any active treatment for performance status/organ impairments, nor invasive nor non-invasive (but high cost) procedures should be recommended. This decision needs to be established case by case and cannot be generalized. There are some situations in which the effectiveness of the cancer treatment is impressive and the toxicity burden relatively low (for example immunotherapy in melanoma), where the treatment can significantly improve survival of patients also in case of extensive organs involvement. Therefore, I do not believe we can extrapolate from this analysis any factor that can drive or help the difficult decision of omission of active treatment, that should be taken with the patient, after extensive discussion, and influenced by many different variables.

I suggest running a median survival for all de novo patients, similarly as it has been done in this paper recently published on JNCI, https://academic.oup.com/jnci/advance-article/doi/10.1093/jnci/djad020/7086066. Another way to improve the quality of the work and the message delivered could be to limit the analysis to a few cancers subtype with a similar prognosis.

6. PLOS authors have the option to publish the peer review history of their article (what does this mean?). If published, this will include your full peer review and any attached files.

Reviewer #1: **Yes: **Luca Giacomelli

Reviewer #2: No

---

## [Author Response · Author response to Decision Letter 0]

20 Jul 2023

Respond to reviewers and to the editor

We thank the reviewers and the editor for the in-depth review of our manuscript and the constructive remarks. Below, please find our response to the specific critiques. 

Reviewer #1

The Authors present the results of a very interesting analysis of the SEER database. The methods are sound and the results - the main of which is, one fifth of de novo diagnosed cancer patients die within the first two months since diagnosis - deserve dissemination.

Reply: We thank the reviewer for the positive comment.

I can offer the following comments:

- the analysis spans from 2004 to 2016. Twelve years in which the early death rate decreased by 4%. Which are Authors' expectations for the last 6 years, 2017-2023, taking into account also the pandemic period?

Reply: We thank the reviewer for raising this point. The observed trend in our analysis is likely to have persisted, given the advancements in targeted therapies and immunotherapy during the study period. The COVID-19 pandemic has led to delayed diagnosis and delayed healthcare seeking behaviors. This factor likely contributes to a dampening effect on the positive trend we would expect.

We think these analyses are of great interest, and should be performed in future research, however the SEER data takes a long time to be fully updated with all variables and is not currently available for the pandemic years. 

We have updated our discussion addressing these valuable comments page 11 lines 237-245. 

- Table 1, the results do not match with the text (e.g., 54.3% of those with early dying appear to be women, but the same figures was reported for men in the text). 

Reply: We are sorry for this mistake, and we thank the reviewer for noticing it. The label in the second line of Table 1 was wrong due to a typing error. We verified the table content again and changed the label to male, accordingly, as mentioned also in the text. 

In the same Table, what does the p-value for race refer to? 

there seem to be major differences in terms of ethnicities in the overall population, but no in the early vs non early mortality groups (e.g., 80.2% of patients with early morality was white, compared with 78.2% in those without early mortality - a difference unlikely to be significant). Can the Authors carefully reconsider all data?

Reply: The p values refer to the results of a Chi square test comparing the distribution of race groups between the early mortality patient and the non-early mortality. We agree the difference is not of major clinical significance, however, due to the large sample size the difference is statistically significant. 

The Authors should always clarify that their findings apply to patients with de novo diagnosed metastatic solid cancer (see for instance the first paragraph of the Discussion, this is not clear).

Reply: We have made the appropriate clarifications throughout the text. Changed in the following places: Abstract, Page 6 line 129, Page 8 line 174, page 8 line 187.

Minor:

- Why does the Introduction start with a word on cancer death rate in the US? (the Authors are from Israel and Cyprus). Can this be generalized?

Reply: We agree with the reviewer’s remark and we changed the introduction accordingly (page 3 lines 43-44).

- Can the Authors expand on the SEER database?

Reply: we added this information in the methods section page 3 lines 66-71.

- in the sentence (Results section) 'Patients that died early were older (70 vs 65), more frequently male (54.3% vs 51.5%), had lower income (56.8% vs 54.9% had an income below $65K a year) and more frequently had brain (15% vs 12.9%), lung (49.2% vs 33.5%) and liver metastasis (33.5% vs 28%).', do the Authors mean 'than those who died after two months since diagnosis'?

Reply: Yes. We updated this sentence that it would read better (page 5 line 116), thanks for the comment.

- Quality of figures should be improved.

Reply: Higher resolution figures were attached.

Reviewer #2: This a retrospective study based on available SEER data, investigating factors of early mortality (2-months) in solid tumors. As a general comment, I believe that the clinical question regarding the drivers of early mortality is crucial, but it is extremely difficult to assess in a cohort of patients with different solid tumors. Indeed, the biology, the response to treatments, the diagnostic process as well as symptoms and possible complication cancer and treatment related significantly differ across cancer subtypes. Therefore, it is difficult to draw generalized conclusions for all solid tumors. In addition, treatment information is missing, and as well as adverse events that may have a role in early mortality. Another missing element that can affect mortality is comorbidities status: patients with a compromised general status or organ impairment due to other diseases are expected to be more likely to rapid decline. An example is COPD in patients with lung cancer: I expect that a patient with an obstructive lung disease and lung cancer is more likely to develop respiratory failure than a patient with normal respiratory function at the time of the cancer diagnosis.

Reply:

Thank you for your valuable comments. We agree that analyzing a diverse cohort of solid tumors presents its unique challenges and requires cautious interpretation of our findings. Nevertheless, the primary objective of our study was to illustrate the prevalence of early mortality across an array of tumors. This comprehensive approach, though not focused on specific tumor types, provides critical insights not only for oncologists but also for general practitioners who frequently encounter patients with suspected metastatic cancer during the diagnostic process. This understanding can guide them in making judicious decisions regarding the utility of certain diagnostic tests, thereby potentially preventing unnecessary procedures. Furthermore, a grasp of early mortality trends across various tumors can improve communication between clinicians, patients, and their families regarding prognosis, which is a crucial aspect of patient care.

With regards to treatment details and adverse events, we fully acknowledge their importance in understanding their impact on early mortality. However, our study's reliance on the Surveillance, Epidemiology, and End Results (SEER) database presented certain limitations, notably the lack of treatment data and adverse events documentation. This constraint prevents us from exploring the potential influences of specific treatments and treatment-induced complications on early mortality. We identify this as a significant gap in our current research and strongly advocate for future investigations that incorporate extensive treatment and adverse event data. This would yield a more comprehensive understanding of early mortality in de-novo metastatic solid cancer patients. Your constructive criticism enhances the discourse surrounding this important topic, and we are appreciative of your inputs. We believe the issue of early mortality is underrecognized and underreported and we hope disseminating our data may provoke further research on the subject. 

In addition, the choice of cut-off of 2 months have created an intrinsic selection bias for aggressive cancer more likely to develop rapid progression. Indeed, authors in the discussion states that 2 months is the usual cut-off to see response in patients in solid tumors. If the baseline is date of biopsy as it seems, this means that patients had the time to probably received 1 cycle of therapy (in average, 1-2 weeks for pathology report, 3-4 weeks to have genomic profile/biological characterization of the tumor). With 2 months as cut-off, many patients with potentially early death are missed. Ideally, for accuracy, the cut-off for early death should be defined according to the cancer subtype.

Reply: We appreciate the critical feedback provided by the reviewer. The primary objective of our study was to shed light on the prevalence of early mortality among patients with metastatic solid tumors. While we acknowledge the potential variation in defining early death within different cancer types, our intention was to investigate the number of individuals across various diagnoses who present for treatment in a critically advanced stage, where the time between diagnosis and death is minimal. By focusing on this aspect, we aimed to highlight the urgency and severity of the situation for these patients. This information is important for all medical practitioners who are involved in the diagnostic process of patients with various cancer types. 

Another issue difficult to overcome in this type of analyses is the following: how is cancer-related death defined? This information should be included in the methods since it is the main objective of the study. In cancer patients, death can be associated with different scenarios including organ failure due to cancer invasion, or secondary consequences of the cancer dissemination (i.e. hypercoagulative state, compression issues etc..). This element is difficult to retrieve from SEER published data.

Reply: Cancer related death is defined as death caused by the specific cancer diagnosis. The cancer registries utilize algorithms to process information from death certificates and identify a single, disease-specific underlying cause of death. 

However, there can be challenges in accurately attributing a single cause of death, leading to potential misattribution. To address this, SEER utilizes cause-specific death classification variables that consider tumor sequence, the site of the original cancer diagnosis, and comorbidities to capture deaths related to the specific cancer even if not explicitly coded as such. This approach leads to a relatively small bias. 

As seen in our data, most patients (>90%) were considered to have died from their cancer but the specific details are not available. Further studies looking at the specific cause of cancer death would be interesting and could highlight specific mechanisms of early mortality in cancer. 

We added this information to the methods section page 4 lines 86-92.

Authors specified that they included only metastatic de novo, but they do not explain why. This choice introduces another selection bias: there is a potential underestimation of early death in those cancer subtype who are less likely to present as de novo (for example, head and neck cancer, that are actually expected to progress rapidly and be lethal for loco-regional complications).

Reply: We specifically chose to concentrate on de novo metastatic cancers to emphasize the significance of the diagnostic process, which is fundamentally different in recurrent cases. The former is usually first seen by non-oncology medical staff whereas the latter are often under oncology follow-up which could introduce bias to the diagnostic process, particularly in cancers that have a higher likelihood of recurrence. We believe our report of early mortality in de novo metastatic cancer patients is of particular importance to primary care physicians and internists who often manage these patients during the diagnostic process. 

The results are consistent with what was expected: higher prevalence of pancreatic cancer and lung cancer, presence of visceral metastases, advanced age, and lower income were associated with early death (the latter element, very relevant form policy standpoint and has not been properly commented in the discussion).

Reply: The significance of our study lies in its ability to shed light on the widespread occurrence of early mortality, the consistency of our results with the expected, strengthens the validity of our analysis. We acknowledge the reviewer's suggestion to place more focus on the lower-income aspect, and in response, we have incorporated this aspect into the discussion section on page 8 lines 185-191. 

 In the discussion section, it is not clear what is the main finding (new finding) and comments on clinical implications are questionable. In my opinion, palliative support needs to be activated in all patients with symptoms, in parallel to the active treatment regardless of if patient it is at risk of early death. I provide a similar argument for invasive diagnostic procedure: for the “primum non nocere” principle, less invasive procedures should be considered for all patients. Liquid biopsy can be diagnostic only for some cancer subtype (i.e. colon cancer, lung cancer) and it is not validated for all type of solid tumors, and in those case in which is validated, it has been performed as standard of care and routinely at the diagnosis for all patients. On the other hand, if a patient is considered ineligible for any active treatment for performance status/organ impairments, nor invasive nor non-invasive (but high cost) procedures should be recommended. This decision needs to be established case by case and cannot be generalized. 

Reply: 

Our study brings to light the alarming prevalence of early mortality among patients diagnosed with de-novo metastatic solid cancer. In our experience, this phenomenon is commonly encountered, particularly in the inpatient hospital setting, but has not been well described. The associated risk factors found were the type of cancer, presence of liver or brain metastases, advanced age, and lower income. These findings emphasize the urgent need for speedier and less invasive diagnostic methods to identify patients who are likely to significantly benefit from specific treatments. We agree that while palliative care activation should indeed be considered for all patients with symptoms, the emphasis in this study is on early mortality and shedding light on this this common clinical scenario increasing discussion on the best approach to these patients. This issue has significant ramifications not only for the patients themselves, but also for the healthcare professionals treating them and the broader healthcare system. 

We aim to identify those who might benefit most from an immediate shift in focus towards end-of-life care rather than pursuing potentially burdensome and less beneficial diagnostic and treatment options.

In our discussion we mention the option of liquid biopsy as a potential alternative in some cases where sequencing is routinely conducted on the tumor tissue after a histological diagnosis (i.e. lung cancer, colon cancer). In our experience, the use of liquid biopsies in histologically undiagnosed cancer patients is currently not routinely done. While this suggestion is not the focus of our study, we mentioned it to stimulate further research on decreasing early mortality. 

There are some situations in which the effectiveness of the cancer treatment is impressive and the toxicity burden relatively low (for example immunotherapy in melanoma), where the treatment can significantly improve survival of patients also in case of extensive organs involvement. Therefore, I do not believe we can extrapolate from this analysis any factor that can drive or help the difficult decision of omission of active treatment, that should be taken with the patient, after extensive discussion, and influenced by many different variables.

I suggest running a median survival for all de novo patients, similarly as it has been done in this paper recently published on JNCI, https://academic.oup.com/jnci/advance-article/doi/10.1093/jnci/djad020/7086066. 

Reply: 

While we have included the median survival statistics above, we are concerned that integrating them into our results may shift the focus of our research away from the high "early mortality" in de novo metastatic cancer. 

Another way to improve the quality of the work and the message delivered could be to limit the analysis to a few cancers subtype with a similar prognosis.

Reply: Thank you for your suggestion. Narrowing down the analysis to cancer subtypes with similar prognosis could help to control for the significant variability in disease trajectories and patient outcomes across different types of cancers. This approach might enable a more detailed examination of the factors associated with early mortality within these specific patient populations.

However, as previously noted the intention of our current study was to provide a broad overview of early mortality across a wide range of de-novo metastatic solid cancers. By doing this, we hoped to highlight the overall scale of the issue and identify some general risk factors. In future research we certainly see the value in conducting more focused analyses as you suggest.

---

## [Decision Letter · Decision Letter 1]

10 Sep 2023

Early death after a diagnosis of metastatic solid cancer – raising awareness and identifying risk factors from the SEER database.

PONE-D-23-02061R1

Dear Dr. Urban,

We’re pleased to inform you that your manuscript has been judged scientifically suitable for publication and will be formally accepted for publication once it meets all outstanding technical requirements.

Kind regards,

Mirosława Püsküllüoğlu, MD, PhD

Academic Editor

PLOS ONE

Additional Editor Comments (optional):

Reviewers' comments:

Reviewer's Responses to Questions

**Comments to the Author**

1. If the authors have adequately addressed your comments raised in a previous round of review and you feel that this manuscript is now acceptable for publication, you may indicate that here to bypass the “Comments to the Author” section, enter your conflict of interest statement in the “Confidential to Editor” section, and submit your "Accept" recommendation.

Reviewer #1: All comments have been addressed

Reviewer #2: All comments have been addressed

2. Is the manuscript technically sound, and do the data support the conclusions?

Reviewer #1: Yes

Reviewer #2: Yes

3. Has the statistical analysis been performed appropriately and rigorously? 

Reviewer #1: Yes

Reviewer #2: Yes

4. Have the authors made all data underlying the findings in their manuscript fully available?

Reviewer #1: Yes

Reviewer #2: Yes

5. Is the manuscript presented in an intelligible fashion and written in standard English?

Reviewer #1: Yes

Reviewer #2: Yes

6. Review Comments to the Author

Reviewer #1: The Authors did a great job in addressing my comments and those of the other reviewer. I think that this paper will be a major contribution to the field.

Reviewer #2: Thanks so much for the detailed response.

I do not have additional comments.

All point have been properly addressed.

The early mortality in solid tumor is an important data that is missing in literature.

7. PLOS authors have the option to publish the peer review history of their article (what does this mean?). If published, this will include your full peer review and any attached files.

Reviewer #1: **Yes: **Luca Giacomelli

Reviewer #2: No

---

## [Editor Report · Acceptance letter]

14 Sep 2023

PONE-D-23-02061R1 

Early death after a diagnosis of metastatic solid cancer – raising awareness and identifying risk factors from the SEER database. 

Dear Dr. Urban:

I'm pleased to inform you that your manuscript has been deemed suitable for publication in PLOS ONE. Congratulations! Your manuscript is now with our production department. 

Kind regards, 

on behalf of

Dr. Mirosława Püsküllüoğlu 

Academic Editor

PLOS ONE